Identification of novel biomarkers in obstructive sleep apnea via integrated bioinformatics analysis and experimental validation

Zhang Kai 1
Wang Caizhen 2
Wu Yunxiao 1
Xu Zhifei 1 zhifeixu@aliyun.com
1 Beijing Children’s Hospital, Department of Respiratory Medicine , Beijing , People’s Republic of China
2 The Second Hospital of Hebei Medical University, Pediatric Intensive Care Unit , Shijiazhuang, Hebei , People’s Republic of China
Loperfido Antonella
Electronic publication date: 2023 Dec 4
Publication date: 2023
Volume: 11
Electronic Location ID: e16608
Received 2023 Jul 28; Accepted 2023 Nov 15
Copyright: © 2023 Zhang et al.
Copyright year: 2023
Copyright holder: Zhang et al.
License: This is an open access article distributed under the terms of the Creative Commons Attribution License, which permits unrestricted use, distribution, reproduction and adaptation in any medium and for any purpose provided that it is properly attributed. For attribution, the original author(s), title, publication source (PeerJ) and either DOI or URL of the article must be cited.
License URL: https://creativecommons.org/licenses/by/4.0/

Keywords: Obstructive sleep apnea, Chronic intermittent hypoxia, Gene expression, Bioinformatics analysis

Funding: Beijing Natural Science Foundation 7212033 National Natural Science Foundation of China 82070092 Respiratory Research Project of National Clinical Research Center for Respiratory Diseases HXZX-20210401 This study was supported by the Beijing the Natural Science Foundation (7212033), the National Natural Science Foundation of China (82070092) and the Respiratory Research Project of National Clinical Research Center for Respiratory Diseases (HXZX-20210401). The funders had no role in study design, data collection and analysis, decision to publish, or preparation of the manuscript.

==============================
Background

Obstructive sleep apnea (OSA) is a complex and multi-gene inherited disease caused by both genetic and environmental factors. However, due to the high cost of diagnosis and complex operation, its clinical application is limited. This study aims to explore potential target genes associated with OSA and establish a corresponding diagnostic model.

Methods

This study used microarray datasets from the Gene Expression Omnibus (GEO) database to identify differentially expressed genes (DEGs) related to OSA and perform functional annotation and pathway analysis. The study employed multi-scale embedded gene co-expression network analysis (MEGENA) combined with least absolute shrinkage and selection operator (LASSO) regression analysis to select hub genes and construct a diagnostic model for OSA. In addition, the study conducted correlation analysis between hub genes and OSA-related genes, immunoinfiltration, gene set enrichment analysis (GSEA), miRNA network analysis, and identified potential transcription factors (TFs) and targeted drugs for hub genes. Finally, the study used chronic intermittent hypoxia (CIH) mouse model to simulate OSA hypoxic conditions and verify the expression of hub genes in CIH mice.

Results

In this study, a total of 401 upregulated genes and 275 downregulated genes were identified, and enrichment analysis revealed that these differentially expressed genes may be associated with pathways such as vasculature development, cellular response to cytokine stimulus, and negative regulation of cell population proliferation. Through MEGENA combined with LASSO regression, seven OSA hub genes were identified, including C12orf54, FOS, GPR1, OR9A4, MYO5B, RAB39B, and KLHL4. The diagnostic model constructed based on these genes showed strong stability. The expression levels of hub genes were significantly correlated with the expression levels of OSA-related genes and mainly acted on pathways such as the JAK/STAT signaling pathway and the cytosolic DNA-sensing pathway. Drug-target predictions for hub genes were made using the Connectivity Map (CMap) database and the Drug-Gene Interaction database (Dgidb), which identified targeted therapeutic drugs for the hub genes. In vivo experiments showed that the hub genes were all decreasing in the OSA mouse model.

Conclusions

This study identified novel biomarkers for OSA and established a reliable diagnostic model. The transcriptional changes identified may help to reveal the pathogenesis, mechanisms, and sequelae of OSA.

Introduction

Obstructive sleep apnea (OSA) is a prevalent sleep-related breathing disorder characterized by the recurrent collapse of the pharynx during sleep, leading to sustained and intense but ineffective breathing efforts. Clinical manifestations of OSA include excessive daytime sleepiness, nocturnal snoring, and repetitive episodes of breathing cessation (Locke, Lee & Sundar, 2022). Long-term untreated OSA can result in intermittent hypoxia, hypercapnia, sleep disturbance, chronic inflammation (such as elevated levels of CRP, IL-6, TNF-α), and stress (Yeghiazarians et al., 2021; Brown et al., 2022). OSA can also cause systemic damage to multiple organs and systems, such as cardiovascular and cerebrovascular diseases, type two diabetes, and cognitive impairment (Malhotra et al., 2021; Gottlieb, 2021). Currently, the prevalence of pediatric OSA ranges from 1.2% to 5.7%, while in the general adult population, the prevalence of OSA falls between 6% and 17%. Among the elderly, this proportion may even reach up to 49%. Therefore, with the increasing aging of the population, its incidence is on the rise (Peppard et al., 2013; Senaratna et al., 2017). This trend may be attributed to factors such as age-related fat accumulation, muscle relaxation, and reduced lung capacity. Currently, polysomnography (PSG) is regarded as the gold standard method for diagnosing OSA, while sleep endoscopy is employed to assess dynamic upper airway obstruction during sleep. Nevertheless, their high cost, intricate procedures, and the limitation in delving into the mechanisms underlying obstructive sleep apnea have drawn significant attention. Therefore, thorough exploration of the etiology and pathogenesis of obstructive sleep apnea, identification of early diagnostic markers, and establishment of treatment objectives become paramount.

According to reports, OSA is a complex, multi-gene genetic disease with a strong genetic influence, as well as environmental factors. The risk of developing OSA increases by more than 1.5 times for first-degree relatives of patients, and approximately 30–40% of variability in the Apnea-Hypopnea Index (AHI) can be explained by genetic factors (de Paula et al., 2016; Mohit, Shrivastava & Chand, 2021; Campos et al., 2020). Twin and pedigree studies have also shown that OSA-related hypoxemia, hypercapnia, obesity, craniofacial skeletal morphology, and other traits are highly heritable, with 30–70% of phenotypic variation determined by shared familial factors (Mukherjee, Saxena & Palmer, 2018; Leader et al., 2021). Currently, high-throughput sequencing of the entire genome DNA microarray has become an effective and relatively economical tool for studying the genetics of complex diseases. Although some OSA molecular biomarkers have been discovered, single genes cannot accurately represent the features of OSA due to its heterogeneity and complex pathophysiological status. Unlike differential gene expression analysis focused on individual genes, Multi-Scale Embedded Gene Co-expression Network Analysis (MEGENA) identifies gene modules at different resolutions through unsupervised clustering and multi-scale clustering analysis (MCA), providing new insights into understanding disease pathogenesis and opportunities for therapeutic intervention. It has been successfully applied to the study of various biological processes such as sleep disorders (Liang et al., 2022), chronic obstructive pulmonary disease (Lin et al., 2022), and thyroid cancer (Xu et al., 2022), and has been shown to be quite effective in identifying candidate biological markers and therapeutic targets.

In this study, we utilized the MEGENA approach to identify genes associated with OSA, and combined the results with least absolute shrinkage and selection operator (LASSO) to select hub genes to construct a prognostic model. Additionally, we conducted an analysis of the biological functions and pathways of these genes to identify potential transcription factors and targeted drugs for the hub genes. To obtain more accurate results, we developed a chronic intermittent hypoxia (CIH) mouse model to simulate the hypoxic conditions during OSA and verified the expression of hub genes in the CIH mouse model (Song et al., 2021). This provides a theoretical basis for identifying potential hub genes related to the pathogenesis of OSA.

Materials

Data download

The dataset for GSE135917, available on the NCBI GEO public database, was obtained by downloading the corresponding Series Matrix File and analysis file (GPL6244). This dataset encompasses expression profiles from a total of 32 patients, divided into 8 cases within the normal group and 24 cases within the disease group. Likewise, the dataset for GSE38792 was acquired from the same NCBI GEO database, comprising the Series Matrix File and analysis file (GPL6244). In this dataset, the expression profiles of 18 patients were included, with eight cases in the normal group and 10 cases in the disease group. GSE135917 was utilized as the foundation for constructing the diagnostic model, serving as the training set. On the other hand, GSE38792 was used to validate the diagnostic model, functioning as the validation set. The workflow of the study is shown in Fig. 1.

Figure 1 Study workflow.

Differential expression analysis

The differential expression analysis of expression profiles, aimed at identifying significantly differentially expressed genes between groups, was performed using the Limma package in the R programming language. Specifically, the Limma package was utilized to investigate the variance in molecular mechanisms associated with sleep apnea by comparing gene expression profiles between control and disease samples. The differential gene screening criteria were set as p value < 0.05 and |log2FC| > 0.585. Subsequently, the obtained differentially expressed genes were visualized through the generation of genetic volcano maps and heatmaps.

Function enrichment analysis

To gain a comprehensive understanding of the functional correlation of the differentially expressed genes, functional annotation was conducted using the Metascape database (www.metascape.org) (Zhou et al., 2019). Specifically, the Metascape database was employed to explore the functional characteristics and associations of these genes. Additionally, Gene Ontology (GO) pathway analysis (Yeh et al., 2003) was performed to gain insights into the specific biological processes and molecular functions associated with the identified genes. For statistical significance, a threshold of Min overlap ≥3 and p ≤ 0.01 was considered as the criteria.

Construction of MEGENA co-expression network

Gene co-expression networks were constructed using Multiscale Embedded Gene Co-expression Network Analysis (MEGENA) (Song & Zhang, 2015). This approach involved initially identifying gene pairs with significant correlations (FDR < 0.05). These meaningful gene pairs were then sorted based on their Pearson correlation coefficients. Subsequently, the sorted gene pairs were tested to determine whether they could be placed on the 3D topological sphere without intersecting with other nodes. This process led to the creation of a co-expression network known as “planar filter networks (PFN),” which belongs to a class of geometric networks that can be embedded on the surface of a sphere without any connecting intersections. To identify network clusters, or gene modules, at different resolutions, unsupervised clustering was performed using multiscale cluster analysis (MCA). MCA effectively split the main module into multiple submodules by searching for and optimizing the partitioning of the module. Finally, genes exhibiting relatively high node values within the key modules were identified as hub genes, indicative of their importance in the network.

Prediction model construction

Following the selection of candidate gene sets, a predictive correlation model was constructed using lasso regression (Friedman, Hastie & Tibshirani, 2010). This modeling technique allowed for the incorporation of the expression values of specific genes. A risk score formula was then developed for each patient, with the weights assigned based on the estimated regression coefficients obtained from the lasso regression analysis. Subsequently, the risk score for each patient was calculated using the constructed formula. To assess the predictive accuracy of the model, the receiver operating characteristic (ROC) curve analysis was performed.

GSEA pathway enrichment analysis

GSEA analysis (Subramanian et al., 2005) is employed to assess the enrichment of a predefined gene set based on the degree of differential expression among two sample types. Genes are sorted according to their expression levels, and the analysis determines whether the preset gene set is enriched at either the top or bottom of the sorted gene list. In our study, GSEA was utilized to investigate the disparities in the KEGG signaling pathway between the high expression group and the low expression group. The aim was to gain insights into the molecular mechanisms associated with the core genes of these two patient groups. The analysis involved 1,000 permutations, with phenotype-based replacements performed during the analysis process.

Cmap drug prediction

The Connectivity Map (CMap) (Lamb et al., 2006) is a gene expression profiling database developed by the Broad Institute, which utilizes intervention gene expression data. Its primary purpose is to unveil the functional associations between small molecule compounds, genes, and disease states. The CMap database encompasses microarray data obtained from five human cell lines, both before and after treatment with 1,309 small molecule drugs. The dataset comprises diverse treatment conditions, encompassing variations in drugs, concentrations, treatment durations, and more. In this study, the differentially expressed genes associated with diseases were leveraged to predict targeted therapeutic drugs for those diseases. By utilizing the CMap database, the aim was to identify small molecule compounds with potential efficacy in modulating the expression profiles of the disease-associated genes.

Drug-gene interaction

The Drug-Gene Interaction Database (DGIdb) (https://www.dgidb.org) (Wagner et al., 2016) is a database of drug-target and gene-phenotype interactions. We searched DGIdb to predict potential drugs or small molecules that interact with hub genes and visualized them using Cytoscape software (Shannon et al., 2003).

Transcriptional regulation analysis of core genes

Cistrome DB (Zheng et al., 2019) is currently one of the most comprehensive databases for studying ChIP-seq and DNase-seq, with a total of 30,451 transcription factor, histone modification, and chromatin accessibility samples from humans and 26,013 from mice. In this study, we explored the regulatory relationships between transcription factors and hub genes through Cistrome DB. The genome files were set to hg38, the transcription start site was set to 10 kb, and the results were visualized using Cytoscape.

Animals and chronic intermittent hypoxia

The research protocol has been approved by the Institutional Ethics Committee of Beijing Children’s Hospital (The approval number: 2020-k-93). The experiment followed guidelines set by the National Institutes of Health. Twelve SPF-grade male C57BL/6J mice aged 6–7 weeks (18–22 g) were obtained from Beijing Vital River Laboratory Animal Technology Co., Ltd., Beijing, China. (License number: SCXK Beijing 2021-0011) and housed under standard conditions (temperature: 21 ± 1 °C; relative humidity: 50 ± 10%) with a 12-h light/dark cycle. After a 1-week adaptation period, the mice (7–8 weeks old) were randomly divided into two groups of six: control, CIH. They were exposed to a chronic intermittent hypoxia (CIH) device for 6 weeks. In brief, the experiment employed a gas control delivery system to alternate the flow of oxygen or nitrogen, circulating the gases between hypoxia and normoxia. The cycle consisted of 8 h per day, with one CIH cycle defined as 90 s, comprising a period of hypoxia (25 s),oxygen increase (20 s), normoxia (25 s), and oxygen decrease (20 s). Throughout the experiment, an oxygen analyzer was used to continuously monitor the concentrations of O2 and CO2. The low oxygen schemes for each group were as follows: control group (21% O2 for 25 s), CIH (5% O2 for 25 s). Except during the CIH period, all animals were housed in an SPF barrier environment and provided with water and food. To ensure that the mice lose consciousness and were sacrificed with minimal pain, we euthanized the mice by rapid neck dislocation, and collected their adipose tissue for further experiments.

RNA extraction and RT-PCR

Firstly, total RNA was extracted from the adipose tissue using Trizol method following the protocol provided by Invitrogen. The purity of RNA was assessed using UV-2000 spectrophotometer (UNIKO, Franksville, WI, USA). Subsequently, cDNA synthesis was performed by reverse transcription. Next, the PCR reaction system was prepared with the following components: 5 μl of 2× SYBR Green Mix, 0.2 μl of forward primer (5 pmol/μl), 0.2 μl of reverse primer (5 pmol/μl), 1 μl of cDNA, and 3.8 μl of ddH2O. The reaction conditions were set as follows: initial denaturation at 95 °C for 3 min, denaturation at 94 °C for 20 s, annealing at 60 °C for 20 s, extension at 72 °C for 40 s, and a total of 40 cycles. The specific primer information can be found in Table S1. Ct values were measured using a fluorescence quantitative PCR instrument (Bio-Rad, Hercules, CA, USA), and the expression levels of the target gene were calculated using the 2-ΔΔCT method.

Statistical analysis

Statistical analyses in this study were conducted using R language (version 4.2.2; R Core Team, 2022). All statistical tests were two-sided, and a significance level of p < 0.05 was used to determine statistical significance.

Results

Differential expression analysis

The GSE135917 dataset was used as the training set, and a total of 676 differential genes were screened, including 401 up-regulated genes and 275 down-regulated genes (Figs. 2A, 2B). We further conducted pathway analysis on this differential gene. Pathway analysis of differential genes was further performed through the Metascape database. The results showed that these differential genes were mainly enriched in pathways such as vascular development, cellular response to cytokine stimulus, and negative regulation of cell population proliferation (Fig. 2C).

Figure 2 Identification of differentially expressed genes in OSA.

(A) Volcano plot of differentially expressed genes, with pink indicating upregulated genes and blue indicating downregulated genes. (B) Heatmap of differentially expressed genes. (C) GO/KEGG enrichment analysis of differentially expressed genes based on the Metascape database, as well as the cluster network composed of enriched pathways, where nodes sharing the same clusters are usually close to each other. (D) MEGENA networks showing the six largest gene modules. Each color represents a module, and triangles represent hub genes in the module.

Construction of MEGENA co-expression network

we utilized a set of 676 differentially expressed genes obtained from the GSE135917 dataset as the candidate gene set for MEGENE (multi-scale embedded gene co-expression network analysis). The initial step involved selecting gene pairs with significant correlation (FDR < 0.05). Subsequently, the gene pairs were sorted based on their correlation coefficients, resulting in the identification of significant gene pairs. After constructing the co-expression network, module clustering was performed to group genes into modules. We identified a total of 26 modules and 1,578 module genes. The largest module, c1_5, consists of 190 genes, followed by c1_6 with 189 genes, c1_15 with 109 genes, and c1_18 with 125 genes. Among them, 29 genes were identified as key genes within the modules (Figs. 2D, S1). These 29 key genes were selected as the final candidate gene set for constructing the LASSO model in subsequent analyses (Figs. 3A, Table S2).

Figure 3 Construction of the diagnostic model.

(A) Five-fold cross-validation for tuning parameter selection in the LASSO model. (B) Distribution of LASSO coefficients for genes. (C) Bar plot of LASSO coefficients for the seven hub genes. (D) Prediction performance on the validation sets.

Prediction model construction

We use the data set GSE135917 as the training set and the data set GSE38792 as the verification set, and select the modeling candidate genes in the previous step to perform feature screening through Lasso regression. The results showed that Lasso regression identified a total of seven genes as the characteristic genes of sleep apnea, which are: C12orf54, FOS, GPR1, OR9A4, MYO5B, RAB39B, KLHL4, as the core genes of follow-up research and construction of prediction models (Fig. 3C). RiskScore = C12orf54 × (−0.17) + FOS × (−0.09) + GPR1 × (−0.06) + OR9A4 × (−0.09) + MYO5B × (−0.05) + RAB39B × (−0.03) + KLHL4 × (−0.006). The results showed that the prediction model built with seven genes had good diagnostic performance, and the area under the AUC curve was one (Fig. S2); we further used the data set GSE38792 as a validation set. The diagnostic model was further verified as an external data set, and the results showed that the model had strong stability, and the area under the curve of GSE38792-AUC was 0.74 (Fig. 3D).

Immune association

We investigated the correlation between the identified seven hub genes and various immune factors sourced from the tumor-immune system interaction (TISIDB) database (Ru et al., 2019). The immune factors encompassed a range of immune-related chemokines, immunosuppressants, immune stimulators, receptors, and more. The obtained results are illustrated (Fig. 4). The analysis conducted in this study provides compelling evidence that the hub genes identified in our study exhibit a strong correlation with the levels of immune cell infiltration. These genes are believed to play crucial roles in shaping the immune microenvironment. The findings suggest their potential significance in modulating immune responses and influencing immune-related processes.

Figure 4 Immune infiltration in OSA.

(A–E) Bubble plot showing Pearson correlation between hub genes and immune factors. (A–E) Represent chemokine, immunoinhibitor, immunostimulator, MHC, and receptor, respectively.

GSEA pathway enrichment analysis

Next, we studied the specific signaling pathways enriched in the seven core genes, and explored the underlying molecular mechanism of the core genes affecting the progression of sleep apnea. The results of GSEA are shown in the figure (Fig. 5). We selected some of the highly significant pathways for centralized display. The C12orf54 gene enriched pathways include the calcium signaling pathway, cytosolic DNA sensing pathway, proteasome and other pathways; the FOS gene enriched pathways include calcium signaling pathway, cytosolic DNA sensing pathway, citrate cycle TCA cycle and other pathways; OR9A4 gene enriched pathways include the cytosolic DNA-sensing pathway, the JAK/STAT signaling pathway, regulation of autophagy and other pathways; GPR1 gene enriched pathways include calcium signaling pathway, cytosolic DNA waysensing pathway, the JAK/STAT signaling pathway and other pathways; KLHL4 gene enriched pathways include the JAK/STAT signaling pathway, the cytosolic DNA-sensing pathway, primary bile acid biosynthesis and other pathways; MYO5B gene enriched pathways include amino sugar and nucleotide sogar metabolism, lyso porphyrin and chlorophyll metabolism and other pathways; RAB39B gene enriched pathways include calcium signaling pathway, cytosolic DNA sensing pathway, drug metabolism other enzymes and other pathways.

Figure 5 GSEA analysis of hub genes.

(A–G) KEGG pathways enriched by hub genes in the high expression group and the low expression group, with the background gene set as KEGG.

miRNA analysis

We screened the characteristics of seven hub genes through the mircode database, and performed reverse prediction on the six genes obtained through screening, and obtained 79 miRNAs, a total of 195 mRNA-miRNA relationship pairs, and visualized them using cytoscape (Fig. 6A).

Figure 6 Correlation analysis of OSA regulatory genes.

(A) miRNA regulatory network of hub genes, where purple represents mRNA and green represents miRNA. (B) Expression differences of OSA disease-regulating genes, where blue represents control patients and pink represents disease patients. (C) Correlation analysis between OSA disease-regulating genes and hub genes, where blue represents negative correlation and red represents positive correlation. (D–G) 2D structure diagram of potential drugs for OSA.

Correlation with obstructive sleep apnea

We obtained 39 genes related to sleep apnea through the GeneCards database (https://www.genecards.org/). Analyze the expression differences of disease genes between groups, and found seven hub genes and ACE, ADIPOQ, AHDC1, BCHE, BDNF, CHAT, CRP, CRY1, CXCL8, EDN1, HTR2A, IL6, LEP, MECP2, PHOX2B, RET, TNF and other genes There are differences in the expression of the two groups of patients.We performed a correlation analysis on hub genes and sleep apnea-related genes. The expression levels of hub genes and sleep apnea-related genes were significantly correlated, among which KILH4 and BCHE were significantly positively correlated (Pearson r = 0.897), OR9A4 Significantly negative correlation with ADIPOQ (Pearson r = −0.762) (Fig. 6C).

Cmap drug prediction

Finally, we took the difference between the two groups of up- and down-regulated mRNAs, and used the Connectivity Map database to predict the drug targeting of the differential genes. The results showed that the expression profiles of calyculin, menadione, KI-8751, LDN-193189 and other drug disturbances and the expression of disease disturbances The spectral negative correlation is more significant, and the drug can alleviate or even reverse the disease state (Figs. 6D–6G).

Predictions of drugs and transcription factors targeting hub genes

DGIdb is used to analyze drugs that may interact with hub genes. Through DGIdb, 10 drugs were found to interact with FOS, which may help develop new targets for treatment (Fig. 7A). In addition, we used seven hub genes as the gene set for this analysis and further explored the transcriptional regulatory network involved in these seven core genes. Using the Cistrome DB online database to predict relevant transcription factors, and using cytoscape for visualization, we constructed a comprehensive transcriptional regulatory network of hub genes in OSA (Fig. 7B).

Figure 7 Targeted drugs and transcription factors of hub genes.

(A) Targeted therapeutic drugs for hub genes, based on the Dgidb database. (B) Transcription factor regulatory network of hub genes.

In vivo experimental validation

RT-PCR experiments were performed on the above seven hub genes using adipose tissue from CIH mice. The hub genes (C12orf54, FOS, GPR1, OR9A4, MYO5B, RAB39B, and KLHL4) were all under-expressed in the CIH group, suggesting that the above mentioned genes do have potential value in the diagnosis of OSA (Fig. 8).

Figure 8 Animal experiment validation.

(A–G) Relative mRNA expression of hub genes (C12orf54, FOS, GPR1, OR9A4, MYO5B, RAB39B, and KLHL4) in adipose tissue.

Discussion

OSA is a clinical syndrome caused by recurrent episodes of breathing pauses or reduced airflow during sleep due to the collapse of the upper airway, leading to a series of pathological and physiological changes. OSA not only causes symptoms such as snoring at night and daytime sleepiness, but also increases the risk of cardiovascular disease, malignant tumors, and even sudden death (Zinchuk & Yaggi, 2020). Currently, the methods for diagnosing OSA include PSG, and for identifying obstructive sites, there is sleep endoscopy. However, these methods require high equipment and personnel demands, are costly, and do not delve into the specific pathogenesis of OSA (Natsky et al., 2022). Therefore, uncovering new molecular mechanisms underlying OSA and identifying potential OSA biomarkers for early diagnosis and targeted treatment are of significant importance. In this study, we used bioinformatics methods to screen the transcriptome data of OSA patients and validated the hub gene by establishing a chronic intermittent hypoxia (CIH) mouse model, and found that some hub genes are related to the clinical features of OSA, providing a new direction for further exploring the pathogenic molecular mechanisms and treatment of OSA. These results suggest that the application of bioinformatics methods in OSA research is promising and can provide a foundation for the development of biomarkers and therapeutic targets for OSA.

Based on GEO microarray dataset, this study compared normal control group with OSA patient group and identified a total of 676 differentially expressed genes. Pathway enrichment analysis revealed that these differentially expressed genes are mainly involved in vasculature development, cellular response to cytokine stimulus, negative regulation of cell population proliferation and other pathways. It was further elucidated that the intermittent hypoxia and hypercapnia caused by OSA can promote the release of inflammatory cytokines from inflammatory cells, leading to systemic inflammatory response, impairment of endothelial repair ability, and promotion of endothelial cell apoptosis (Zdravkovic et al., 2022; Gottlieb, 2021). In addition, hypoxia can also cause peripheral red blood cell increase, platelet aggregation, hemodynamic changes, promote thrombus and plaque formation, and promote vascular remodeling, ultimately leading to the development of cardiovascular disease (Fernandez-Bello et al., 2022; Chen, Lin & Zeng, 2021). These findings reveal the pathogenesis of OSA, particularly the damage to the vascular system and the inflammatory response, providing new directions for the early diagnosis and treatment of OSA. We hope that these results will contribute to a better understanding of the pathogenesis of OSA and provide new insights into the development of biomarkers and therapeutic targets for OSA.

Furthermore, this study utilized the MEGENA combined with LASSO method to screen seven crucial genes, namely C12orf54, FOS, GPR1, OR9A4, MYO5B, RAB39B, and KLHL4. These genes are all downregulated genes, positively correlated with known OSA-related genes BCHE, BDNF, CRP, among others, and negatively correlated with ADIPOQ, LEP, and others. Among them, GPR1, OR9A4, and KLHL4 mainly function in the JAK/STAT signaling pathway. Currently, it is widely believed that the JAK/STAT signaling pathway can regulate biological processes such as cell proliferation, differentiation, and apoptosis, and participate in various biological functions of cytokines, such as inducing the secretion of inflammatory factors, aggravating inflammatory reactions, lowering vascular relaxation, leading to endothelial dysfunction, and endothelial cell apoptosis (Hu et al., 2021; Xin et al., 2020). Therefore, literature has reported the close correlation of abnormal activation of the JAK/STAT signaling pathway with the occurrence and development of various diseases, such as atherosclerosis, myocardial ischemia-reperfusion, nephritis (Cai et al., 2021), and arthritis (Simon et al., 2021), but there are few reports on its relationship with OSA. C12orf54, FOS, and RAB39B mainly participate in intracellular membrane transport, signal transduction, and may be associated with OSA-induced cell apoptosis, endothelial dysfunction, and cardiovascular disease occurrence (Herr, Stettner & Kubin, 2013; Gambarte et al., 2019). The discovery of these screened hub genes provides a new direction for the study of the molecular mechanism of OSA. Further research on the specific role of these genes in the pathophysiological process of OSA may help discover new therapeutic targets and strategies, improving the diagnostic and treatment level of OSA.

In the immune system, biological processes such as inflammation and signal transduction interact with each other, jointly maintaining the stability of immune function and the balance of the internal environment. Therefore, the inflammation induced by OSA may also have a synergistic effect on immune response. To further understand the relationship between immune response and OSA, we analyzed the correlation between the seven hub genes and immune factors such as chemokines, immunosuppressants, immune stimulants, and receptors. The results showed that these hub genes are closely related to the level of immune cell infiltration and play an important role in the immune microenvironment. Further research on the specific roles of these genes in the immune system will help to deepen our understanding of the pathogenesis of OSA, discover new therapeutic targets and strategies, and improve the diagnosis and treatment of OSA.

The expression trends of C12orf54, FOS, GPR1, OR9A4, MYO5B, RAB39B, and KLHL4 in animal experiments were consistent with the results of the training set and the testing set. Furthermore, we conducted an additional analysis of the transcription factors associated with hub genes and potential targeted drugs to explore new avenues for treating OSA. Studies suggest that FOS may be a potential target gene for treating OSA. In our data set and animal experiments, FOS was the only hub gene that exhibited significant differences and shared common transcription factor regulation with other hub genes. In animal experiments, the expression level of FOS was positively correlated with disease, indicating that FOS may play an important role in the pathogenesis of OSA. Therefore, further research and analysis of the specific role of FOS in OSA is necessary. Additionally, we identified some potential targeted drugs that may have significant therapeutic effects for OSA. These findings provide new directions and insights for the treatment of OSA.

This study represents the first application of MEGENA methodology to investigate the role of differentially expressed genes in the pathogenesis of OSA, validated through in vivo animal experiments. Nevertheless, our study has some limitations. Firstly, due to the fact that the data we used were obtained from adipose tissue, further biomarkers in the blood of OSA patients are desired for clinical application. Secondly, although we have validated the hub gene in animal experiments, the validity of the hub gene for OSA diagnosis must be confirmed in as many datasets as possible in the future.. In addition, due to the unavailability of detailed clinical data, we were unable to evaluate the relationship between patients with different clinical symptoms and complications and hub genes. Future research could expand the sample size and perform detailed phenotyping of patients, to better understand the potential mechanisms underlying OSA. We hope that future research can overcome these limitations and delve deeper into the pathogenesis of OSA, providing more scientific evidence for the development of better treatment options.

Conclusion

In this study, we have identified C12orf54, FOS, GPR1, OR9A4, MYO5B, RAB39B, and KLHL4 as hub genes that are differentially expressed in OSA patients compared to healthy individuals. These genes are involved in biological processes related to inflammation activation, intracellular membrane transport, and signal transduction pathways, and could potentially serve as biomarkers for OSA diagnosis and treatment. Additionally, we have found relationships between these genes and immune factors, suggesting that immune infiltration also plays an important role in OSA pathology, which may help to further elucidate the mechanisms of OSA. Future studies can further validate the biological functions and mechanisms of these hub genes to better understand their role in OSA.

Supplemental Information

Supplemental Information 1 Code.

Click here for additional data file.

Supplemental Information 2 RT-PCR.

Click here for additional data file.

Supplemental Information 3 MEGENA network of OSA genes.

MEGENA co-expression network of differentially expressed genes, with larger nodes representing higher numbers of genes.

Click here for additional data file.

Supplemental Information 4 The ROC curve of the training set.

The predictive performance on the training set.

Click here for additional data file.

Supplemental Information 5 Primer sequences in this study.

Click here for additional data file.

Supplemental Information 6 Clinical characteristics of the patient.

Click here for additional data file.

Supplemental Information 7 29 characterized genes from MEGENA analysis.

Click here for additional data file.

Supplemental Information 8 RiskScore coefficients and formulas for each gene.

Click here for additional data file.

Supplemental Information 9 Author Checklist.

Click here for additional data file.

Additional Information and Declarations

Competing Interests

Author Contributions

Animal Ethics

Data Availability

The authors declare that they have no competing interests.

Kai Zhang conceived and designed the experiments, performed the experiments, analyzed the data, prepared figures and/or tables, and approved the final draft.

Caizhen Wang performed the experiments, analyzed the data, prepared figures and/or tables, and approved the final draft.

Yunxiao Wu analyzed the data, prepared figures and/or tables, authored or reviewed drafts of the article, and approved the final draft.

Zhifei Xu conceived and designed the experiments, authored or reviewed drafts of the article, and approved the final draft.

The following information was supplied relating to ethical approvals (i.e., approving body and any reference numbers):

Institutional Ethics Committee of Beijing Children’s Hospital provided full approval for this research (2020-k-93)

The following information was supplied regarding data availability:

The data is available at NCBI GEO: GSE135917, GSE38792. The raw measurements are available in the Supplemental Files.

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
