# Peer review of "Identification of novel biomarkers in obstructive sleep apnea via integrated bioinformatics analysis and experimental validation"

_PeerJ, doi:10.7717/peerj.16608_

## Round 0.1 · original submission · Major Revisions

Dear Author,

I recommend to revise the article according to the Reviewers' indications.

·

Basic reporting

This article is written in in clear English completely comprehensible by foreigners .
A little more reference could be added on OSA genetics, in particular evidence at line 66 (de Paula, L.K., Alvim, R.O., Pedrosa, R.P., Horimoto, A.R., Krieger, J.E., Oliveira, C.M., Pereira, A.C., and Lorenzi- 424 Filho, G. 2016. Heritability of OSA in a Rural Population. CHEST 149:92-97. 10.1378/chest.15-0843) is obtained by subtraction of other factors and it is not strong although this confirms the novelty of this article.
All data, raw and analysis are provided as well as ethical committee approval
Results are self contained and supported by in silico and in vivo experiments

Experimental design

This article defines correctly the aim and delivers clearly conclusions.
All procedures and study materials as well as algorithms used are consultable and clear, granting reproducibility,
Investigation is performed with an high technical standard and, for what I can infer, ethical standard.
The novelty of this study lies in the lack of research on genetic basis of OSAS, this can be a good waypoint

Validity of the findings

Data are sound and robust, conclusions well explained.
My only concern is the ability to translate this kind of research in real knowledge: OSAS is a multifactorial disease and thus genes involved can play the role of confoundments: a gene involved in hunger mechanism can indirectly affect fat deposits nearby airways and thus correlations may appear stronger than in clinical reality

Additional comments

I reccomend this article for its replicable and clear methods, I like how they are exploring an uncharted and promising field and thus I hope that more studies could follow and dissipiate my concerns about realiability real life

Reviewer 2 ·

Basic reporting

The manuscript titled "Identification of Novel Biomarkers in Obstructive Sleep Apnea via Integrated Bioinformatics Analysis and Experimental Validation" presents a commendable effort to explore the genetic basis of obstructive sleep apnea (OSA) and identify potential biomarkers. While the introduction succinctly provides the context and importance of OSA, there are several areas that require clarification and enhancement to strengthen the manuscript.The introduction appropriately introduces the clinical manifestations of OSA and the challenges associated with its diagnosis. However, the concept of "chronic inflammation" as a hallmark of OSA is somewhat oversimplified. The term "chronic inflammation" is quite nonspecific and needs further elaboration. Furthermore, the manuscript mentions the increasing prevalence of OSA, particularly among the aging population, but it fails to clarify that the incidence of OSA varies across different populations. Providing a brief overview of the epidemiology of OSA, including factors influencing its incidence, would add depth to the understanding of the disease's impact.

Experimental design

The introduction outlines the application of Multi-scale Embedded Gene co-expression Network Analysis (MEGENA) to identify OSA-associated genes. However, it lacks a critical aspect of clinical evaluation. Describing the characteristics of the patients enrolled, including clinical parameters and severity of OSA, would lend credibility to the study and enhance the relevance of the findings to the real-world patient population.
To further clarify, it would be helpful to know more about the characteristics of the patients within each group. Specifically:
• Age and Sex: What are the age ranges and sex distributions of the patients in each group? Understanding the age and sex distribution can help contextualize the findings and determine if there are any potential age- or sex-related differences in gene expression and AHI levels.
• AHI Levels: Are the patients' AHI levels available or documented? The AHI is a key parameter used to diagnose the severity of obstructive sleep apnea. Knowing the AHI levels of the patients in both the normal and disease groups will be crucial for correlating gene expression patterns with disease severity.
• Clinical Characteristics: Are there any additional clinical characteristics or health parameters that are available for these patients? Factors such as BMI (Body Mass Index), comorbidities, and medical history can influence the gene expression patterns and the overall understanding of OSA.
• Ethnicity and Demographics: Additionally, information about the ethnic and demographic backgrounds of the patients would contribute to the comprehensive interpretation of the results, as these factors can play a role in genetic predisposition and disease manifestation.

Validity of the findings

The manuscript emphasizes the limitations of polysomnography (PSG) as a diagnostic tool for OSA, but it neglects to acknowledge the role of sleep endoscopy and the alternative diagnostic approaches. Incorporating sleep endoscopy alongside PSG in the diagnostic discussion would offer a more comprehensive view of the available diagnostic techniques.
The assertion that PSG is "high cost and complexity" is not entirely accurate. The manuscript states that PSG's cost and complexity limit its application in clinical settings. However, it's important to note that high-throughput genetic arrays are also resource-intensive. It would be helpful to provide additional reasons why genetic analysis is useful beyond cost-effectiveness, such as its potential to uncover novel molecular mechanisms underlying OSA.

Reviewer 3 ·

Basic reporting

Zhang et al utilize microarray data to identify and validate seven hub genes that are differentially expressed between healthy and diseased obstructive sleep apnea (OSA) patients. They conduct differential analysis, gene co-expression analysis, lasso regression, functional enrichment, and pathway analysis. Overall, the authors provide a comprehensive literature review on OSA and provide an external and an experimental validation of the discovered hub genes, proposing a potential set of diagnostic biomarkers for OSA.

1. Please include a brief description of the code or add comment in the .R file for each task.
2. In addition to website, please cite the references for the software/databases: Metascape (Zhou et al., https://pubmed.ncbi.nlm.nih.gov/30944313/), GO, MEGENA, lasso, GSEA, CMap, Cistrome, Cytoscape, TISIDB, DGIdb, etc.

Experimental design

One major concern with the Methods section: There are several inconsistencies between the code and written methods/procedure. Please go through each section in details and make sure all procedures are described correctly.

In the Materials section:
1. Please use FDR method for multiple testing correction to select differentially expressed genes (e.g., use FDR<0.05 or FDR<0.1 or use different thresholds). Current strategy described in Methods line 109-110 does not properly control the false discovery rate. By checking the supplemental code (line 23), it was specified as adjustP=0.05. But in the Material section, p-value cutoff is used. Please make sure the code and the procedure are consistent.
1. Please describe why to use |logFC|>0.585 as cutoff. This fold change cutoff seems very subjective. Please either use a quantile-based method to choose cutoff or use a more common cutoff (e.g., 1) for downstream analysis.
2. For functional enrichment analysis, please include the result for using adjusted p-value cutoff. Also, please provide the rationale for using min overlap >3.
3. In prediction model construction section, please state clearly that the prediction accuracy/auc is evaluated on the validation set. Please include cross validation in this section as well.
4. In GSEA pathway analysis, please describe how to determine statistically significantly enriched pathways (criteria).
5. In statistical analysis section, please either use same FDR cutoff throughout all analyses, or summarize the different p-value/fdr criteria for different tasks.

Validity of the findings

In the Results section:
1. Fig. 2A: For the volcano plot, the X axis is labelled as log2FC but in Material section, it’s logFC. Please make sure it’s consistent. Please highlight important genes that will be discussed on this volcano plot. Also, please incorporate previous suggestions to regenerate the results.
2. Fig. 2B: Please add a legend for red-blue color in the heatmap. (e.g., scaled gene expression)
3. Fig. 2C: Please check the figure panel, it’s marked as A/B. Also, please include a higher resolution for the network plot.
4. Fig. 2D: Please consider using an alternative plot to present the main result. Current figure is hard to interpret. It's hard to see the difference of the node size. Supplemental Fig1B-E contain more biologically meaningful findings and could be considered to present one of the results in this figure panel.
5. Sup Fig1A: Please consider using different color to highlight the modules that will be discussed – c1_4, c1_5, c1_6, c1_15, c1_16, c1_18.
6. Line 224-226: Please describe about specific modules that were selected (e.g., c1_4, c1_5, c1_6, c1_15, c1_16, c1_18).
7. Line 227: please briefly describe the definition the class ‘c’ (e.g., how to determine the child module).
8. Line 229: It was not clear how to determine the final set of 29 genes. In SupFig1B-E, there are child hub (diamond), hub genes (triangle) and unmarked genes (circle). Which 29 genes are the final set? Please highlight those gene in different colors. Please document the final gene list in a supplemental table. Please use a specific example and describe the procedure in plain English.
9. By checking the supplemental code (line 321), it was specified as nfold=5. But in the results section, 10-fold cross validation is used. Please make sure the code and the procedure are consistent.
10. Fig3D: the prediction performance for training set can be included as supplemental figure.
11. Line238-241: Please do not include the long decimals in the main text. Would suggest using a supplemental table to document the coefficients and include the formula in the supplemental.
12. Line 248: Please include full name of TISIDB in its first appearance.
13. Fig5: Please provide a summary figure to summarize the pathways that are shared across different hub gene enriched pathways. For example, calcium signaling pathway is significant in C12orf54, FOS, GPR1, RAB39B enriched pathways.
14. Fig6B: multiple testing (FDR) should be used for statistical significance (*, **).
15. Fig6C: the pearson correlation is a measurement of linear correlation and thus using a line rather than spline may be more appropriate. Please describe the testing procedure (p-value) in this figure.

---

## Round 0.2 · accepted · Accept

Based on the reviewers' recommendations, the article is accepted.

Reviewer 2 ·

Basic reporting

No comment

Experimental design

No comment

Validity of the findings

No comment

Additional comments

The study is very interesting and explores new aspects and possibilities in the management of a widely prevalent condition like obstructive sleep apnea, which affects millions of people. Diagnostic and therapeutic management is still varied, and the results of this study, if further investigated with a larger number of patients in other studies, could provide a significant contribution to the scientific community.

Reviewer 3 ·

Basic reporting

no comment

Experimental design

no comment

Validity of the findings

no comment